# The Resistance of Seven Host Plants to *Tetranychus merganser* Boudreaux (Acari: Tetranychidae)

**DOI:** 10.3390/insects13020167

**Published:** 2022-02-04

**Authors:** Guadalupe Treviño-Barbosa, Salvador Ordaz-Silva, Griselda Gaona-García, Agustín Hernández-Juárez, Sandra Grisell Mora-Ravelo, Julio César Chacón Hernández

**Affiliations:** 1Instituto de Ecología Aplicada, Universidad Autónoma de Tamaulipas, División del Golfo 356, Colonia Libertad, 87019 Victoria City, Tamaulipas, Mexico; gtbarbosa@uat.edu.mx (G.T.-B.); ggaona@docentes.uat.edu.mx (G.G.-G.); sgmora@docentes.uat.edu.mx (S.G.M.-R.); 2Facultad de Ingeniería y Negocios San Quintin, Universidad Autónoma de Baja California, Carretera Ensenada-San Quintín, Km 180.2, Ejido Padre Kino, 22930 San Quintin, Baja California, Mexico; salvador.ordaz.silva@uabc.edu.mx; 3Departamento de Parasitología, Universidad Autónoma Agraria Antonio Narro, Calzada Antonio Narro 1923, Buenavista, 25315 Saltillo, Coahuila, Mexico; chinoahj14@hotmail.com

**Keywords:** red spider mite, host plants, demographic parameters, damage, oviposition, egg hatch

## Abstract

**Simple Summary:**

The red spider mite, *Tetranychus merganser* is one of the most economically important pests in papaya and prickle pear cactus cultivars, causing major damage to fruit and defoliation. In recent years, *T. merganser* has increased the number of its host plants. The mechanisms of resistance of a plant to herbivorous arthropod include antixenosis and antibiosis. Antixenosis refers to the plant mechanism to affect feeding and oviposition of arthropods; antibiosis refers to the plant capacity to affect the biology of the arthropod. The aim of this research is to assess antibiosis and antixenosis as resistance mechanisms in seven host plants (*Thevetia ahouai*, *Carica papaya*, *Phaseolus vulgaris*, *Moringa oleifera*, *Pittosporum tobira*, *Helietta parvifolia*, *Capsicum annuum* var. *glabriusculum*) to red spider mites. Oviposition and damage by feeding of *T. merganser* were greater on *C. papaya* than on the other host plants. The population growth of the spider mite was lower in *P. tobira* and *T. ahouai* than in the other host plants. Results based on the analysis of demographic parameters, food intake, survival and oviposition of *T. merganser* females suggest that *P. tobira* and *T. ahouai* were the most resistant to red spider mites, whereas *C. papaya* was the most susceptible of the seven host plants. The resistant plants can be studied as alternatives in the management of red spider mites.

**Abstract:**

Red spider mites, *Tetranychus merganser* Boudreaux (Acari: Tetranychidae), is an agricultural pest that causes economic losses in papaya and nopal crops in Mexico. The aim of this research was to assess antibiosis and antixenosis as resistance mechanisms in seven host plants (*Thevetia ahouai*, *Carica papaya*, *Phaseolus vulgaris*, *Moringa oleifera*, *Pittosporum tobira*, *Helietta parvifolia*, *Capsicum annuum* var. *glabriusculum*) to red spider mites. Antixenosis was evaluated by non-preference for oviposition and feeding, antibiosis by infinitesimal rate of increase, finite rate of increase and doubling time, and the percentage of spider mites mortality. Oviposition and damage by feeding of *T. merganser* were significantly greater on *C. papaya* than on the other host plants. The growth rate of the spider mite was significantly lower in *P. tobira* and *T. ahouai* than in the other host plants. The percentage of hatched eggs of *T. merganser* was significantly higher in *P. vulgaris* than in the other plant species. Based on the demographic parameters, survival, food intake, and oviposition, these results indicated that compared with *C. papaya*, *P. tobira* and *T. ahouai* were more resistant. These results may be due to the fact that they were plants species of different families. The resistant plants can be studied as alternatives in the management of *T. merganser*.

## 1. Introduction

Tetranychidae family include more than 1300 species of phytophagous mites, of which one hundred can be considered pests, ten of them being of great importance [1]. The mites that cause serious damage to crops and ornamental plants are found in the genera *Tetranychus* Dufour, *Panonychus* Yokoyama, *Oligonychus* Berlese, and *Eutetranychus* Banks [1,2]. The red spider mite, *Tetranychus merganser* Boudreaux (Acari: Tetranychidae), causes severe damage by its feeding in different species of plants of the family Aquifoliaceae, Apocynaceae, Cactaceae, Caricaceae, Cucurbitaceae, Euphorbiaceae, Fabaceae, Moringaceae, Oleaceae, Pittosporaceae, Rosaceae, Ranunculaceae Rutaceae and Solanaceae [1,3,4,5,6,7,8]. The red spider mite is distributed in the United States, China, Mexico, and Thailand [1]. Moreover, it is considered a potential pest for Mexican agriculture [3,8], e.g., causing losses of 586 ± 234 dollars per hectare in prickly pear cactus, (*Opuntia ficus-indica* L.) Miller (Cactaceae) crops [3]. The red spider mite can develop and reproduce in a wide range of climatic factors [9,10]. Ullah et al. [9] evaluated the behavior of *T. merganser* at different temperatures, 15 to 37.5 °C and 60–70% relative humidity, on bean disc, *Phaseolus vulgaris* L. (Fabaceae). They documented that *T. merganser* has better performance at 30 °C. Furthermore, Reyes-Pérez [10] found that optimal development for spider mites was at 27 °C on *Carica papaya* L. (Caricaceae), when evaluated between 19 and 35 °C and 60 ± 2% relative humidity. Chacón-Hernández et al. [11] reported that the performance of spider mites was better on beans when compared to wild chili peppers, *Capsicum annuum* L. var. *glabriusculum* (Solanaceae). The population growth parameters of *T. merganser*, such as daily egg production, survival, food intake, and rate of development, may vary in response to changes in temperature, host plant species, and nutrition quality of plants [9,10,11].

*T. merganser* is controlled through the use of insecticides and acaricide chemicals. However, the red spider mites’ short life cycle and high reproductive potential allows them to quickly develop resistance to these compounds [12]. The use of botanical extracts [13] and predatory mites, mainly from the family Phytoseiidae [4], are more effective and sustainable strategies to the management of red spider mites, with the added benefit of causing minimal impact on the environment. In Mexico, Chacón-Hernández et al. [11] observed that wild chili pepper (*C. annuum* var. *glabriusculum* (Dunal) Heiser & Pickersgill) was resistant to *T. merganser*. Plant responses to herbivorous arthropod attack are based on genetically inherited qualities, generally divided into three categories: antixenosis, antibiosis and tolerance [14,15,16]. Antibiosis occurs when a phytophagous arthropod is negatively affected, especially in its biology, by chemical and morphological presents in resistant host plants. Antixenosis or deterrence is the non-preference of a phytophagous arthropod to a resistant plant and denotes the anti-feeding and anti-oviposition caused by biophysical or allelochemical factors, resulting in the late acceptance or absolute rejection of a plant as a host. Tolerance is a polygenic trait that allows a plant to resist, repair or recover from damage caused by the phytophagous arthropod [14,15,16,17,18]. Polyphagous arthropods have the ability to tolerate or resist the defense mechanisms of host plants, which allows them to feed and reproduce [19]. The aim of this research is to assess antibiosis and antixenosis as resistance mechanisms in seven host plants species (*Thevetia ahouai* (L.) A. DC. (Apocynaceae), *C. papaya* L., *P. vulgaris* L., *Moringa oleifera* Lam. (Moringaceae), *Pittosporum tobira* (Thunb.) W.T. Aiton (Pittosporaceae), *Helietta parvifolia* (Gray) Benth. (Rutaceae), *C. annuum* L. var. *glabriusculum*) to *T. merganser*.

## 2. Materials and Methods

### 2.1. Red Spider Mite Colony

A red spider mite colony was started with biological material obtained from the Population Ecology Laboratory, Institute of Applied Ecology, Autonomous University of Tamaulipas (IEA-UAT). To increase the spider mite population, female and male *T. merganser* were placed on bean plants (*P. vulgaris*) under greenhouse conditions at 30 ± 2 °C and 70 ± 10% relative humidity (RH).

### 2.2. Collection and Preparation of Plant Material

For this study, seven plant species reported as hosts of *T. merganser* were selected [5,6,7,8,10] (Table 1).

We collected 20 mature leaves of each species of host plants in its natural habitat (Table 1). The leaves of each host plants were transported in resealable plastic bags inside a cooler with frozen gel packs at a temperature of 5 ± 2 °C to the Population Ecology Laboratory at the Ecology Institute of the Autonomous University of Tamaulipas. The transfer time of the leaves to the laboratory depended on the location of the host plants, but was between 15 to 30 min. From the 20 leaves, we selected three clean leaves, without the presence of any damage and without symptoms of the presence of fungi or bacteria. The leaves were treated with a 2 min wash with sodium hypochlorite solution with 1.5% and had 2 × 2 cm squares cut out from each leaf.

### 2.3. Experimental Design

We used the sand technique described by Ahmadi [19] with modifications. We cut leaf squares of each species of host plants, seven in total. The squares measured 2 × 2 cm and were cut with the help of a sterile scalpel. We placed the leaf squares on cotton soaked in water with the underside side up. We placed each leaf square inside a 5 cm diameter Petri dish, with 10 *T. merganser* females per leaf square of each host plant. Previously, ten adult females and five males of *T. merganser* were placed in each leaf square of the plant to improve reproduction and oviposition. After 24 h, the males were removed, leaving only the females (the observed eggs were removed). The experiment was carried out under laboratory conditions at 28 ± 1 °C and 70–80% relative humidity (RH), with a photoperiod of 12:12 h (light: dark). We randomly assigned three leaf squares to seven groups, one group for each host plant species. The leaf squares of each host plant species were the replicas, and had three replicas per group, 21 in total. Adult females *T. merganser* were two days old when transferred to the leaf square.

### 2.4. Antixenosis 

Antixenosis was determined by non-preference (oviposition and feeding) of *T. merganser* for any host plant. The feeding of *T. merganser* was visually estimated on each leaf square through the damage index proposed by Nachman and Zemek [20], where 0 = 0% damage (no feeding damage) and 5 = 81% to 100% damage by feeding (a dense mark or wilt caused by spider mite feeding of the leaf square). Only one person registered the number of eggs laid per female and the percentage of feeding damage at 24, 48, 72 and 96 h with the help of a dissecting microscope (UNICO Stereo and Zoom Microscopes ZM180, Princenton, NJ, USA), to avoid bias during the evaluation.

### 2.5. Antibiosis

We used the infinitesimal rate of increase (r, day^−1^), the finite rate of increase (λ), and the population doubling time (D_T_) to determine antibiosis, and r was calculated by:r = (1/t) × ln(N_t_/N_0_)(1)
where N_t_ is the number of individuals at time t (surviving adult females plus the eggs laid at the end of the bioassay), and N_0_ is the number of individuals at time 0 (initial cohort = 10 adult females of *T. merganser*) and t is the number of days elapsed from the start to the end of the bioassay (equal to 3 days).

The finite growth rate, i.e., the number of times the population multiplied in a unit of time, was calculated as:λ= antilog_e_ r(2)

The doubling time in which the population doubled was calculated by [21]:D_T_ = Ln(2)/r(3)

The demographic parameters (r, λ and D_T_) indicate the population growth of the red spider mite.

### 2.6. Mortality

We used the Chacón-Hernández et al. [11] formula to measure the percentage of mortality of *T. merganser*. We measured mortality by the average percent of dead individuals (drowned) outside the leaf square (Σd_i_/n) × 100, where d_i_ is the number of drowned individuals and n the number of individuals on the leaf square [11]. We recorded the number of dead and alive mites at 24, 48, 72 and 96 h.

### 2.7. Hatched Eggs of Tetranychus merganser

On the fifth day, after leaving the females alone, the number of immature mites that emerged from the eggs laid by *T. merganser* female was recorded. The time it takes to hatch a *T. merganser* egg is around 3.6 ± 0.3 days, at 27.5 °C in bean discs (*P. vulgaris*), with a photoperiod of 16: 8 h light: dark and 60–70% HR [9]. Meanwhile, hatching time on papaya discs (*C. papaya*) is around 4.10 ± 0.52 days, at 27.0 °C with a photoperiod of 14:10 h L: D and 60 ± 2% RH [8]. Based on literature, eggs that did not hatch during five days were considered nonviable due to natural death or will take longer to hatch.

### 2.8. Statistic Analysis

The number of laid eggs, dead mites, live mites and the percentage of damage by feeding were registered daily during four days. These data was studied using analysis of variance of repeated measurements (ANOVArm). The number of immature mites was recorded on the fifth day and the demographic parameters (r, λ and D_T_) were calculated on the fourth day. These data was analyzed using one-way ANOVA, and in both cases, means were separated by Tukey’s multiple range comparison test (*p* ≤ 0.05). Finally, we correlated the mean number of eggs laid with average percent of live females, mean damage with average percent of live females and mean damage with average of eggs laid per of *T. merganser* female. The SAS/STAT software was used for all analyzes [22].

## 3. Results

### 3.1. Antixenosis

The number of *T. merganser* eggs laid per female differed significantly between host plants (F = 352.64; df = 6, 14; *p* < 0.0001), among observation time (F = 46.09; df = 3, 42; *p* < 0.0001) and host×time interaction (F = 19.30; df = 18, 42; *p* < 0.0001). The number of eggs laid on *C. papaya* was significantly higher while on *P. tobira* and *T. ahouai* were significantly lower (Tukey’s test, *p* < 0.05) (Table 2), this suggests *P. tobira* and *T. ahouai* were most resistant to *T. merganser*. Moreover, we found a positive correlation between fecundity and host plant species (Figure 1A).

The feeding damage of *T. merganser* differed significantly between host plants (F = 26.19; df = 6, 14; *p* < 0.0001), both in relation to time (F = 473.13; df = 3, 42; *p* < 0.0001) and host×time interaction (F = 2.31; df = 18, 42; *p* = 0.0129). Damage was significantly greater in *C. papaya* and *P. vulgaris*, while in *H. parvifolia*, *P. tobira* and *T. ahouai*, it was significantly less (Table 3), which indicates *H. parvifolia*, *P. tobira* and *T. ahouai* were more resistant to *T. merganser*. Furthermore, we found a positive correlation between feeding damage and host plant species of red spider mites (Figure 1B), and among feeding damage and average of eggs laid per each *T. merganser* female (Figure 1C).

### 3.2. Antibiosis

Infinitesimal rate of increase (r), finite rate of increase (λ), and doubling time (D_T_) of *T. merganser* were significantly different between host plants (F = 350.35; df = 6,14; *p* < 0.0001; F = 380.61; df = 6, 14; *p* < 0.0001; F = 222.09; df = 6,14; *p* < 0.0001), respectively. The highest average (±SE) of the r of *T. merganser* was observed in *C. papaya* (0.8482 ± 0.01) and the lowest r in *P. tobira* (0.3379 ± 0.00) and *T. ahouai* (0.3421 ± 0.01). The average number of mites added to the population per female per day (λ) was higher in the leaves of *C. papaya* (2.3356 ± 0.01) and lower in *P. tobira* (1.4021 ± 0.01) and *T. ahouai* (1.4080 ± 0.01). The average time in which the spider mite population doubled its population (D_T_) were greater in *P. tobira* (5.9212 ± 0.10) and *T. ahouai* (5.8532 ± 0.14), and less pronounced in *C. papaya* (2.3577 ± 0.01) (Table 4), which indicated *P. tobira* and *T. ahouai* more resistant to the development of *T. merganser*.

### 3.3. Mortality

The number of dead red spider mites differed significantly between host plants (F = 9.53; df = 6, 14; *p* < 0.0003), among observation time (F = 47.00; df = 3, 42; *p* < 0.0001) and host × time interaction (F = 2.58; df = 18, 42; *p* = 0.0058). In general, the largest percentage of dead mites was observed on *H. parvifolia* and *T. ahouai*, which suggests these plant species are more resistant to *T. merganser* (Table 5). 

### 3.4. Hatched Eggs of Tetranychus merganser

The number of eggs hatched on the fifth day differed significantly amongst host plants (F = 123.03; df = 6, 14; *p* < 0.0001). The percentage of hatched eggs was significantly higher on *P. vulgaris* than on *H. parvifolia* (Table 6). This indicates that the host plant species has an effect on the eggs laid during the first 24 h, which are supposed to hatch on the fifth day after laying.

## 4. Discussion

Oviposition is the first stage in the arthropod life cycle; it becomes exposed to the environment, and it is the most important event in the chain of interactions between herbivores and plants [23,24,25]. The number of eggs laid by *T. merganser* females differed amongst host plants, suggesting that plants influence red spider mite biology. In this study, the oviposition trend of *T. merganser* in relation to the host plants was: *C. papaya* > *P. vulgaris* > *M. oleifera* > *C. annuum* var. *glabriusculum* > *H. parvifolia* ≥ *P. tobira* ≥ *T. ahouai*. This indicates that the mite prefers to oviposit on caricaceous plants than on fabaceous, moringaceous, solanaceous, rutaceous, pittosporaceous and apocynaceous plants. Several studies have shown that *Tetranychus* spp. fecundity is related to the host plant species. Yano et al. [26] found a positive correlation between the mean number of eggs laid in 5 days and the host plant acceptance of *T. urticae*. They evaluated seven herbaceous plants (*Houttuynia cordata* (Saururaceae), *Rumex crispus* (Polygonaceae), *Solidago altissima* (Asteraceae), *Cayratia japonica* (Vitaceae), *Desmodium* sp. (Fabaceae), *Rorippa indica* (Brassicaceae) and *Taraxacum officinale* (Asteraceae)) and four cultivated ones (*Fragaria* (strawberry) sp. (Rosaceae), *Chrysanthemum* sp. (Asteraceae), *P. lunatus* (Fabaceae) and *P. vulgaris* (Fabaceae)), and found that *T. urticae* Koch lays more eggs on fabaceous plants. Further, De Lima et al. [27] documented that the total fecundity and daily oviposition rate of *T. bastosi* Tuttle, Baker & Sales were higher in *C. papaya* (50.6 ± 4.4), than on *P. vulgaris* (36.1 ± 2.5) and *Manihot esculenta* (Euphorbiaceae) (26.5 ± 2.2). Chacón-Hernández et al. [11] reported that *T. merganser* oviposited more on fabaceous plants than on solonaceous plants. Islam et al. [28] found that the fecundity of *T. truncatus* Ehara was greater on malvaceous plants (*Corchorus capsularis* L. (Malvaceae) (129.6 ± 3.95)) followed by fabaceous plants (*Lablab purpureus* L. (86.5 ± 3.08)) and caricaceous plants (*C. papaya*, 84.2 ± 3.59). Puspitarini et al. [29] documented the highest number of eggs laid per day and a higher total fecundity rate of *T. urticae* on both caricaceous and roseaceous plants compared to asteraceous plants. Greco et al. [30] found the mean of eggs per female per 5 days of *T. urticae* was lower on amaryllidaceous (*Allium cepa* L. and *A. porrum* L.) and apiaceous plants (*Petroselinum sativum* (Mill.) FUSS) than on rosaceous plants (*Fragaria ananassa* var. Selva Duch.), while Ullah et al. [9] reported a higher number of eggs per female red spider mite (75.91 ± 1.24 ≈ 15.18 eggs/female/day) on bean leaf discs during the first five days oviposition at 30 °C, RH 60–70% and photoperiod of L16: D8 h. Reyes-Pérez et al. [10] reported lower oviposition per *T. merganser* female (6.95 eggs/day) on *C. papaya* discs at 23 °C, photoperiod of 14:10 h light: dark and relative humidity of 60 ± 2%. It is likely that *T. merganser* females lay a higher numbers of eggs on *C. papaya* than on other plant species, this is caused by differences in nutrient contents, morphological characteristics and different secondary metabolites found in different plant families. [26,27,28,29,30].

The food intake was evaluated by the percentage of damage, which was significantly different between host plants. *H. parvifolia*, *P. tobira* and *T. ahouai* were the host plants that suffered less damage from the feeding of the red spider mite, which suggests that those three plants species possibly have different morphological characteristics or secondary metabolites that deter the feeding of *T. merganser*. Secondary metabolites like alkaloids, flavonoids, terpenes and phenols are produced by many plants species and are stored in the cell walls of leaves to deter feeding and oviposition of arthropods [15]. The diversity and quantity of these metabolites in plants affect herbivory species (e.g., oviposition, mate selection, and feeding); and are generally due to both genetic and environmental influences [16,31,32]. In this study, the plant species differ greatly in their suitability as hosts for *T. merganser* when measured in terms of fecundity and as resources for feeding. Studies suggest that a protein present in caricaceous, roseaceous, and asteraceous plants can increase the fecundity of *T. urticae* [29]. In this regard, Islam et al. [28] mention that this occurs because adult females need feeding resources (e.g., nitrogen and carbohydrate) to develop mature ovaries and eggs and obtain energy. 

The demographic parameters measured in this experiment (r, λ and D_T_) indicate that *T. merganser* population growth differ between host plants species. These parameters demonstrated that *T. merganser* had the best performance on *C. papaya*. This is mainly due to higher egg production (9.39 eggs per 4 days) and low mortality of mites. Chacón-Hernández et al. [11] reported values of r of *T. merganser* on fabaceous (0.4237) and solanaceous (0.6014) plants, lower than those obtained in this present study (0.7133 and 0.5568, respectively). De Lima et al. [27] found that *T. bastosi* had the best reproduction rates on both caricaceous and fabaceous plants than on euphorbiaceous plants. There are a variety of factors influencing the population growth parameters of tetraniquids, e.g., host plant species, quality of the host plant, strain of mite, environmental factors, plant breeding method and data analysis method [14,26,27,29,33].

In this research, the percentage of eggs hatched on the fifth day after laying differed between host plants, although more in-depth research is required to understand these differences. Plants initiate the attack on arthropods when they lay their eggs on the leaves [34,35]. Hilker and Fatouros [24] and Beyaert et al. [36] mention that the laying of eggs by herbivorous arthropods induces secondary metabolites in the plant that can prevent the hatching of the eggs or can generate a detachment of the part of the leaf where the egg was laid by the herbivore. Hilker and Meiners [37] mentioned that high concentrations of volatile terpenes could enter the egg via the aeropyles in the outermost layer, i.e., the protein chorion and expand through the wax layer, thus reaching the embryo behind the vitelline envelope and serosa which would cause the death of the egg.

## 5. Conclusions

To conclude, the results show that the host plants *H. parvifolia*, *P. tobira* and *T. ahouai* present antibiosis and antixenosis on the red spider mite, causing a lower oviposition and feeding of *T. merganser*, a low growth rate in its population and a higher period of time to double the population of red spider mites. Therefore, these plant species can be studied as alternatives in the management of *T. merganser*.

## Figures and Tables

**Figure 1 insects-13-00167-f001:**
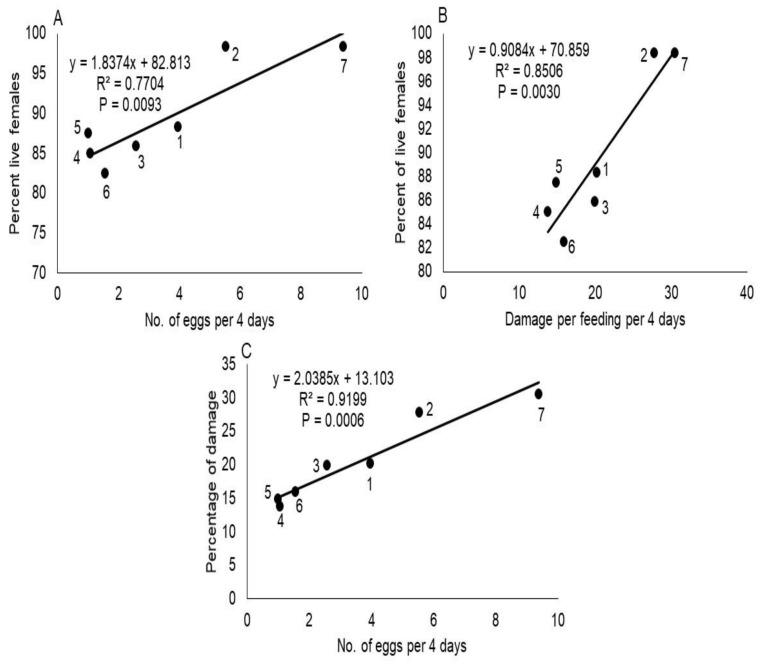
The correlation between mean number of eggs laid with percent of live females (**A**), damage with percent of live females (**B**) and damage with mean number of eggs produced by these mites in 4 days (**C**), on different hosts plants. The plots in the figure represent different plant species: (1) *Moringa oleifera*, (2) *Phaseolus vulgaris*, (3) *Capsicum annuum* var. *glabriusculum*, (4) *Thevetia ahouai*, (5) *Pittosporum tobira*, (6) *Helietta parvifolia*, and (7) *Carica papaya*.

**Table 1 insects-13-00167-t001:** Host plants used for the study of resistance to *Tetranychus merganser*.

Family	Scientific Name	Common Name	Location	Coordinates *	MASL	Reported as Host Plant by
Apocynaceae	*Thevetia ahouai* (L.) A. DC.	Acotope	Victoria City (Urban area)	23°46′10.49″ N 99°9′44.95″ W	308	Monjarás-Barrera et al. [6]
Caricaceae	*Carica papaya* L.	Papaya	Victoria City (Semi-urban area)	23°46′22.3″ N 99 5′58.5″ W	256	Reyes-Pérez et al. [10]
Fabaceae	*Phaseolus vulgaris* L.	Bean	Victoria City (Urban area)	23°45′28.84″ N 99°9′53.54″ W	297	
Moringaceae	*Moringa oleifera* Lam.	Moringa	Victoria City	23°46′22.8″ N 99°5′57.1″ W	256	Monjarás-Barrera et al. [5]
Pittosporaceae	*Pittosporum tobira* (Thunb.) W.T. Aiton	Chinese orange blossom	Victoria City (Urban area)	23°42′54″ N 99°10′48″ W	448	Monjarás-Barrera et al. [7]
Rutaceae	*Helietta parvifolia* (Gray) Benth.	Barreta	Peregrina Canyon in Protected Natural Area “Altas Cumbres”, Victoria City	23°46′41″ N 99°12′12″ W	365	Monjarás-Barrera et al. [7]
Solanaceae	*Capsicum annuum* L. var. *glabriusculum* (Dunal) Heiser y Pickersgill	Chile piquin	Protected Natural Area “Altas Cumbres”, Victoria City	23 41′52″ N 99°11′04″ W	411	Monjarás-Barrera et al. [8]

MASL: meters above sea level. * Location of the plant host for this research work.

**Table 2 insects-13-00167-t002:** Number mean of eggs laid per female of *Tetranychus merganser* on different species of host plants.

Host Plant	Number of Eggs *				
	24 h	48 h	72 h	96 h	General Average
*Carica papaya*	11.00 ± 0.58 a	9.70 ± 0.16 a	9.39 ± 0.27 a	7.46 ± 0.18 a	9.39 ± 0.41 a
*Phaseolus vulgaris*	7.33 ± 0.44 b	5.63 ± 0.35 b	4.77 ± 0.20 b	4.41 ± 0.22 b	5.54 ± 0.37 b
*Moringa oleifera*	4.59 ± 0.30 c	4.07 ± 0.29 c	3.63 ± 0.12 c	3.57 ± 0.12 c	3.97 ± 0.16 c
*Capsicum annuum* var. *glabriusculum*	2.17 ± 0.19 d	2.20 ± 0.18 d	3.03 ± 0.19 c	2.92 ± 0.12 d	2.58 ± 0.14 d
*Helietta parvifolia*	1.72 ± 0.12 d	1.56 ± 0.12 de	1.51 ± 0.09 d	1.49 ± 0.04 e	1.57 ± 0.05 e
*Pittosporum tobira*	1.15 ± 0.10 d	1.02 ± 0.06 e	1.04 ± 0.07 d	1.10 ± 0.09 e	1.00 ± 0.04 f
*Thevetia ahouai*	1.00 ± 0.16 d	1.00 ± 0.08 e	0.98 ± 0.04 d	0.97 ± 0.09 e	1.06 ± 0.05 f
General Average	4.14 ± 0.79 A	3.60 ± 0.66 B	3.48 ± 0.62 B	3.13 ± 0.48 C	

* Mean values and ± standard error (SE) within columns and rows followed by different lowercase and uppercase letters, respectively, are significantly different (*p* < 0.05; ANOVArm and Tukey’s HSD).

**Table 3 insects-13-00167-t003:** Percentage of feeding damage of *Tetranychus merganser* in different host plants.

Host Plant	Percentage of Damage *				
	24 h	48 h	72 h	96 h	General Average
*Carica papaya*	16.00 ± 2.08 a	24.67 ± 2.67 a	34.33 ± 2.33 a	47.00 ± 0.73 a	30.50 ± 3.60 a
*Phaseolus vulgaris*	14.67 ± 1.20 ab	24.67 ± 1.45 a	31.00 ± 2.08 ab	40.67 ± 1.76 ab	27.75 ± 2.94 b
*Moringa oleifera*	9.67 ± 0.88 bcd	15.33 ± 1.45 b	23.67 ± 0.67 bc	32.33 ± 1.76 bc	20.25 ± 2.64 c
*Capsicum annuum* var. *glabriusculum*	10.00 ± 1.15 bc	15.67 ± 2.03 b	23.33 ± 1.86 bc	30.67 ± 1.20 c	19.92 ± 2.45 c
*Helietta parvifolia*	6.67 ± 0.88 cd	10.33 ± 0.88 b	19.67 ± 1.45 c	27.00 ± 2.52 c	15.92 ± 2.49 d
*Pittosporum tobira*	6.00 ± 0.58 cd	10.33 ± 1.45 b	16.67 ± 1.67 c	26.33 ± 3.18 c	14.83 ± 2.45 d
*Thevetia ahouai*	4.33 ± 0.67 d	9.33 ± 0.67 b	17.00 ± 1.15 c	24.33 ± 0.67 c	13.75 ± 2.32 d
General Average	9.62 ± 0.98 D	15.76 ± 1.46 C	23.67 ± 1.52 B	32.62 ± 1.83 A	

* Mean values and ± standard error (SE) within columns and rows followed by different lowercase and uppercase letters, respectively, are significantly different (*p* < 0.05; ANOVArm and Tukey’s HSD).

**Table 4 insects-13-00167-t004:** Demographic parameters of *Tetranychus merganser* on different host plants.

Host Plant	Demographic Parameters *		
	Growth Rate	Finite Growth Rate	Doubling Time
*Carica papaya*	0.8482 ± 0.00 a	2.3356 ± 0.01 a	2.3577 ± 0.01 e
*Phaseolus vulgaris*	0.7133 ± 0.01 b	2.0409 ± 0.02 b	2.8045 ± 0.03 de
*Moringa oleifera*	0.6146 ± 0.01 c	1.8491 ± 0.01 c	3.2548 ± 0.04 cd
*Capsicum annuum* var. *glabriusculum*	0.5568 ± 0.02 d	1.7557 ± 0.03 d	3.6006 ± 0.12 c
*Helietta parvifolia*	0.4068 ± 0.01 e	1.5022 ± 0.02 e	4.9242 ± 0.14 b
*Pittosporum tobira*	0.3379 ± 0.01 f	1.4021 ± 0.01 f	5.9212 ± 0.10 a
*Thevetia ahouai*	0.3421 ± 0.01 f	1.4080 ± 0.01 f	5.8532 ± 0.14 a

* Mean values and ± standard error (SE) are presented. Different letters indicate significant differences (*p* < 0.05; ANOVA and Tukey’s HSD).

**Table 5 insects-13-00167-t005:** Percentage of mortality and survival of *Tetranychus merganser* on leaf squares of different host plants at different times.

Host Plant	Mortality *				
	24 h	48 h	72 h	96 h	General Average
*Carica papaya*	0.00 ± 0.00 a	0.00 ± 0.00 b	3.33 ± 3.33 bc	3.33 ± 3.33 c	1.67 ± 1.12 b
*Phaseolus vulgaris*	0.00 ± 0.00 a	0.00 ± 0.00 b	0.00 ± 0.00 c	6.67 ± 3.33 bc	1.67 ± 1.12 b
*Moringa oleifera*	6.67 ± 3.33 a	6.67 ± 3.33 ab	10.00 ± 5.77 abc	23.33 ± 3.33 ab	11.67 ± 2.71 a
*Capsicum annuum* var. *glabriusculum*	0.00 ± 0.00 a	13.33 ± 3.33 a	16.67 ± 3.33 ab	26.67 ± 3.33 a	14.17 ± 3.13 a
*Helietta parvifolia*	6.67 ± 3.33 a	13.33 ± 3.33 a	23.33 ± 3.33 a	26.67 ± 3.33a	17.50 ± 2.79 a
*Pittosporum tobira*	3.33 ± 3.33 a	6.67 ± 3.33 ab	20.00 ± 0.00 a	20.00 ± 5.57abc	12.50 ± 2.79 a
*Thevetia ahouai*	6.67 ± 3.33 a	10.00 ± 0.00 ab	16.67 ± 3.33 ab	26.67 ± 3.33a	15.00 ± 2.61 a
General Average	3.33 ± 1.05 D	7.14 ± 1.40 C	12.86 ± 2.09 B	19.05 ± 2.38 A	
	Survival				
*Carica papaya*	100.00 ± 0.00 a	100.00 ± 0.00 a	96.67 ± 3.33 ab	96.67 ± 3.33 a	98.33 ± 1.12 b
*Phaseolus vulgaris*	100.00 ± 0.00 a	100.00 ± 0.00 a	100.00 ± 0.00 a	93.33 ± 3.33 ab	98.33 ± 1.12 b
*Moringa oleifera*	93.33 ± 3.33 a	93.33 ± 3.33 ab	90.00 ± 5.77 abc	76.67 ± 3.33 bc	88.33 ± 2.71 a
*Capsicum annuum* var. *glabriusculum*	100.00 ± 0.00 a	83.67 ± 3.33 b	83.33 ± 3.33 bc	73.33 ± 3.33 c	85.83 ± 3.13 a
*Helietta parvifolia*	93.33 ± 3.33 a	83.67 ± 3.33 b	76.67 ± 3.33 c	73.33 ± 3.33 c	82.50 ± 2.79 a
*Pittosporum tobira*	96.67 ± 3.33 a	93.33 ± 3.33 a	80.00 ± 0.00 c	80.00 ± 5.77 abc	87.50 ± 2.79 a
*Thevetia ahouai*	93.33 ± 3.33 a	90.00 ± 0.00 ab	83.33 ± 3.33 bc	73.33 ± 3.33 c	85.00 ± 2.61 a
General Average	96.67 ± 1.05 A	92.86 ± 1.40 B	87.14 ± 2.09 C	80.95 ± 2.38 D	

* Mean values and ± standard error (SE) within columns and rows followed by different lowercase and uppercase letters, respectively, are significantly different (*p* < 0.05; ANOVArm and Tukey’s HSD).

**Table 6 insects-13-00167-t006:** Average number (±SE) of hatched eggs laid by *Tetranychus merganser* during the first 24 h.

Host Plant	Hatched Eggs *	Percentage of Eggs Hatched
*Carica papaya*	91.00 ± 5.86 a	82.62 ± 1.36 abc
*Phaseolus vulgaris*	71.67 ± 4.67 b	97.67 ± 0.55 a
*Moringa oleifera*	38.00 ± 1.73 c	89.17 ± 4.05 ab
*Capsicum annuum* var. *glabriusculum*	19.00 ± 1.53 d	87.84 ± 2.36 ab
*Helietta parvifolia*	10.67 ± 1.20 d	66.32 ± 5.12 c
*Pittosporum tobira*	7.33 ± 0.88 d	76.21 ± 7.63 bc
*Thevetia ahouai*	8.67 ± 0.88 d	81.56 ± 4.22 abc

* Mean values and ± standard error (SE) are presented. Different letters indicate significant differences (*p* < 0.05; ANOVA and Tukey’s HSD).

## Data Availability

The data presented in this study are available on request from the corresponding author.

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
