# Peer review of "The Resistance of Seven Host Plants to *Tetranychus merganser* Boudreaux (Acari: Tetranychidae)"

_insects, 2022, doi:10.3390/insects13020167_

Round 1
Reviewer 1 Report
The maunscript entitled ‘Host plants resistance to Tetranychus merganser Boudreaux (Acari: Tetranychidae)’ is to investigate the resistance of seven host plants to T. merganser by measurement of antixenosis and antibiosis. The manuscript is well written in English. I have some major and minor concerns about this study as following:
Major:
- 1) The title is a little bit exaggerating, because seven host plants could not be the representatives of all hosts. The title could be modified as ‘The resistance of seven host plants to Tetranychus merganser Boudreaux (Acari: Tetranychidae)’.
- 2) Current study is designed to contribute to the management of merganser by applying resistant plants, so the authors studied how host plants affect T. merganser by recording of antixenosis and antibiosis. However, the authors paid quite a lot attention to the susceptibility of the host plants in the section of results and discussion. My suggestion is to rewrite this part and stay focusing on the host plants resistance across the entire manuscript.
Minors:
Line 40, change ‘indicate’ to ‘indicated’.
Line 40, ‘P. tobira and T. ahouai were more resistant and C. papaya more susceptible’ seems unprofessional. It is better to changed as ‘compared with C. papaya, P. tobira and T. ahouai were more resistant.’.
Line 47 and 54, add ‘as’ behinde of ‘considered’ .
Line 68, change ‘therefore’ to ‘so’.
Line 111, change ‘female’ to ‘females’.
Line 163, what is ‘gl’?
In table 2 and 3, their sub-titles are same listed as 24h, 48h, 72h and 96h. Table 2 showed the mean number of eggs laid per 24 hours, while Table 3 present a cumulative damage during the time of 24 hours, 48 hours, 72 hours and 96 hours. It would be confusing the readers.
Reviewer 2 Report
##Manuscript title: Host plant resistance to Tetranychus merganser Boudreaux (Acari: Tetranychidae)
##General comment
In this paper, the effect of seven hosts on demographic parameters and biology of Tetranychus merganser was evaluated. The authors reported variations on these aspects according to the host, which can be reliable information to the management of T. merganser. However, the paper's central message is unclear and flawed by some points.
First, in the introduction, the authors did not provide arguments to validate the work; instead, they offered general concepts about plant defenses without a precise aim of the study. Second, there are concerns regarding the experimental design, the number of replicates, repellency bioassay, and data analysis. Further, the discussion section does not provide arguments to validate the results, as the authors mainly use results from other studies with general speculation as a conclusion. In addition, they tried to justify the research on the argument of environmental variations where species came from, but it is related to botany family as the species used were different (not same species from the different environments). I have provided examples of these shortcomings below and hope they can be helpful to the authors. But, in the current form, I do not recommend this paper for publication.
#Introduction
The authors did not provide arguments to validate the work; instead, they offered general concepts about plant defenses (lines 58-66), which is not a novelty to justify the research. Also, we did not read information about pest performance on the families reported in lines 51-54, which appeared only in the discussion (lines 230-246) section. The authors keep in mind that we (readers) need to know the points to justify the work performed and the new research advances.
Lines 50-54: Is there related literature on how the species perform these families? If so, provide a background.
Lines 56-58: It is not limited to plants grown in natural habitats. For instance, plants in greenhouses can suffer daily variations in temperature. Please revise.
#Material and methods
The authors should clarify three main points:
- The experimental design is not straightforward. In lines 109-111, it is not clear if the authors inserted three-leaf squares of each host plant or if one leaf from each host was inserted in each group. It is mainly due to the initial statement `the leaf squares of each host plant were randomly divided into seven groups. Please clarify.
- This effect was not tested in the repellency bioassay since the authors inserted females in the leaf square. How can volatiles from the host plant repel the insect when they are already on? The argument of `drowned or became trapped in cotton` is weak because it can occur by spatial competition or random behavior. It could be an irritability effect, but I`m unsure if the methodology can support it.
- In the statistic analysis, the authors did not consider the repeated measures performed; instead, they analyzed data considering each evaluation time (e.g., lines 161-163). In my view, this is not correct since each evaluation is related to the previous one and performed in the same leaf square. Also, in data analyzed as percentual, the initial number of individuals (table 5) and eggs laid (table 6) were not equal in the replicates, then it should be included as weights in models because the authors used proportions as dependent variables (Bates et al. 2015).
_Bates D, Maechler M, Bolker B & Walker S. 2015. Fitting linear mixed-effects models using lme4. Journal of Statistical Software 6, 1–48.
Lines 109-111: Why do the authors use only three replicates? Is it enough to estimate variance among treatments?
Lines 115-118: How many people performed the analysis? As the scale depends on the observer, it should be mentioned to disclose potential bias during the evaluations.
Line 119: Can the authors justify the hours as a factor? In my view, there is no argument to support it.
Lines 149-151: How about the natural mortality of eggs? The authors did not provide any morphological or chemical analysis to support this statement.
Line 154: larvae?
#Results
Would you please revise the section after the comments that I have made on material and methods?
Table 3: Number of eggs? I guess it is feeding damage.
Table 4: The demographic parameters are written in Spanish.
Line 205: It was not supported by the experiments. There is only a cue regarding the palatability of host plants.
Table 5: Fix ‘supervival’.
Table 6: Italicized the mite species name in the table title.
#Discussion
This section does not provide arguments to validate the results. The author mainly uses results from other studies with general speculation. For instance, in lines 230-245, they report five studies and then in lines 246-249 conclude general assumptions that were not tested in the paper, like leaf nutritional quality, morphological barriers, and allelochemicals. My suggestions are: (i) the authors used species from different families, then discuss the results based on this topic, not the environment where plants came from; (ii) provide other results to support the arguments, not rewrite them, they are already published!
Lines 219-224: The authors did not test it, and arguments here are circular. They used different species from different areas. Therefore, the variations are related to species families rather than the environment where they come from. To validate the argument, the authors could have experimented with the same species but from different areas. Please rewrite.
Author Response
Please see the attachment
Regards
Sincerely,
Julio Cesar Chacón-Hernández
Corresponding author

Reviewer 3 Report
Dear Authors
This study is aimed to assess antibiosis and antixenosis as a resistance mechanisms in seven host plants toward red spider mites, and the results showed that Pittosporum tobira and Thevetia ahouai were more resistant while Carica papaya is more susceptible.
some comments:-
The introduction is short and provides some background information, however, there is a wealth of literature describing studies on Tetranychus spp. and their host plants, and the ability to resist, if you can add some of them will be great.
62-64 Add references
104 Is there a starvation period before the experiments?
224- Check the reference style, and check along with the text.
313- Please re-check all reference styles, for example (Year position, Bold (case)…), (see some online insects paper).
Author Response
Please see the attachment.
Regards,
Sincerely,
Julio Cesar Chacón-Hernández
Corresponding author

Round 2
Reviewer 2 Report
Manuscript ID insects-1528939
Title: Resistance of seven host plants to Tetranychus merganser Boudreaux (Acari: Tetranychidae)
The authors addressed my concerns raised in the first review and also improved the manuscript. The following comments and suggestions are of a minor nature and could be considered in a minor revision of the manuscript.
L. 25: Please replace supervival by survival.
L. 149: Please replace use by used.
In tables 2, 3 and 5: Please provide an additional footnote regarding the uppercase and lowercase letters. Plus, I would suggest keeping a, b, c, order instead of w, z, y because its kind of difficult to follow them.
Author Response
Dear reviewer, we make all the suggestions made by you.
|
reviewer comments |
Reply |
|
L. 25: Please replace supervival by survival. |
Suggestion accepted, "supervival" is replaced by "survival" |
|
L. 149: Please replace use by used. |
Suggestion accepted, "use" is replaced by "used" |
|
In tables 2, 3 and 5: Please provide an additional footnote regarding the uppercase and lowercase letters. Plus, I would suggest keeping a, b, c, order instead of w, z, y because its kind of difficult to follow them. |
Suggestion accepted. The footnote of tables 2, 3, and 5 were replaced by “Mean values and ± standard error (SE) within columns and rows followed by different lowercase and uppercase letters, respectively, are significantly different (P <0.05; ANOVArm and Tukey's HSD). |
